# Emerging Fusarium Mycotoxins Fusaproliferin, Beauvericin, Enniatins, and Moniliformin in Serbian Maize

**DOI:** 10.3390/toxins11060357

**Published:** 2019-06-19

**Authors:** Igor Jajić, Tatjana Dudaš, Saša Krstović, Rudolf Krska, Michael Sulyok, Ferenc Bagi, Zagorka Savić, Darko Guljaš, Aleksandra Stankov

**Affiliations:** 1Faculty of Agriculture, University of Novi Sad, 21000 Novi Sad, Serbia; igor.jajic@stocarstvo.edu.rs (I.J.); sasa.krstovic@stocarstvo.edu.rs (S.K.); bagifer@polj.uns.ac.rs (F.B.); zagorka.savic@polj.uns.ac.rs (Z.S.); darko.guljas@stocarstvo.edu.rs (D.G.); aleksandra.stankov@polj.uns.ac.rs (A.S.); 2Institute of Bioanalytics and Agro-Metabolomics, Department IFA-Tulln, University of Natural Resources and Life Sciences Vienna (BOKU), A-3430 Tulln, Austria; rudolf.krska@boku.ac.at (R.K.); michael.sulyok@boku.ac.at (M.S.); 3Institute for Global Food Security, School of Biological Sciences, Queens University Belfast, University Road, Belfast BT7 1NN, UK

**Keywords:** emerging mycotoxins, *Fusarium*, LC-MS/MS, maize, Serbia

## Abstract

Emerging mycotoxins such as moniliformin (MON), enniatins (ENs), beauvericin (BEA), and fusaproliferin (FUS) may contaminate maize and negatively influence the yield and quality of grain. The aim of this study was to determine the content of emerging *Fusarium* mycotoxins in Serbian maize from the 2016, 2017, and 2018 harvests. A total of 190 samples from commercial maize production operations in Serbia were analyzed for the presence of MON, ENs, BEA, and FUS using liquid chromatography-tandem mass spectrometry (LC-MS/MS). The obtained results were interpreted together with weather data from each year. MON, BEA, and FUS were major contaminants, while other emerging mycotoxins were not detected or were found in fewer samples (<20%). Overall contamination was highest in 2016 when MON and BEA were found in 50–80% of samples. In 2017 and 2018, high levels of MON, FUS, and BEA were detected in regions with high precipitation and warm weather during the silking phase of maize (July and the beginning of August), when the plants are most susceptible to *Fusarium* infections. Since environmental conditions in Serbia are favorable for the occurrence of mycotoxigenic fungi, monitoring *Fusarium* toxins is essential for the production of safe food and feed.

## 1. Introduction

Mycotoxins have become one of the most important food contaminants in modern society. They are toxic secondary metabolites that are usually produced by *Aspergillus*, *Penicillium*, and *Fusarium* fungi in favorable environmental conditions. Among these species, *Fusarium* are the most prevalent mycotoxin-producing fungi in the northern temperate regions, and mainly Central and SoutheasternEurope [1]. *Fusarium* molds are known as producers of several mycotoxins, including both “traditional” as trichothecenes, zearalenone, and the fumonisins [2] and “emerging toxins” as moniliformin (MON), enniatins (ENs), beauvericin (BEA), and fusaproliferin (FUS) [3]. Additionally, Kovalsky et al. [3] found a co-occurrence of “traditional” toxins with EN, MON, and BEA. The main producers of emerging *Fusarium* mycotoxins in cereals are *Fusarium avenacum*, *Fusarium verticillioides (moniliforme)*, *Fusarium proliferatum,* and *Fusarium subglutinans* [1].

MON is a sodium or potassium salt of 1-hydroxycyclobut-1-ene-3,4-dione [4]. It was discovered by Cole et al. [5] while screening for toxigenic products of *F. verticillioides* isolated from southern leaf blight-damaged maize seed. BEA and ENs belong to a group of cyclodepsipeptidesthat may have antibiotic, insecticidal, and cytotoxic effects [6,7,8,9]. FUS is a toxic sesterterpene originally isolated by Ritieni et al. [10] from *F. proliferatum* in autoclaved maize cultures.

BEA is known as a cholesterol acyltransferase inhibitor [11] and it is toxic to several human cell lines [12,13]. Additionally, BEA can induce apoptosis and DNA fragmentation [14]. FUS is a sesterterpene identified from maize cultures of *F. proliferatum* isolated from maize [15]. FUS is toxic to *Artemiasalina*, to the lepidopteran cell line SF-9 and to the human nonneoplastic B-lymphocyte cell line IARC/LCL 171 [16]. FUS can induce teratogenic effects in chicken embryos [17]. Kriek et al. [18] found that ducklings and rats fed diets containing MON led to muscular weakness, respiratory distress, cyanosis, coma, and death.

There is limited data on the toxicity and occurrence of “emerging” mycotoxins. These mycotoxins are neither routinely determined nor legislatively regulated. Their presence has been reported in cereals from several countries [19,20,21,22,23]. In a recent EFSA report [24], an opinion on the presence of ENNs and BEA in food and feed was made, but the lack of relevant toxicity data did not a risk assessment. Currently, maximum levels for emerging *Fusarium* mycotoxins are not been regulated.

Maize is one of the most susceptible cereals to the presence of *Fusarium* molds. Infection of maize may lead to grain size and protein decreasing as well as harming germination. The final result is a decrease in yield and feed quality. Additionally, a consequential mycotoxin production is another highly problematic outcome of *Fusarium* infection.

In Serbia, arable land covers approximately 75.5% of utilized agricultural land. In the structure of sown arable land areas, cereals comprised 67.9%, industrial crops comprised 15.7%, vegetables comprised 2.6%, and fodder crops comprised 9.1% in 2016 [25]. However, cereals were grown on 1,763,575 ha in 2016, which is lower compared to 2015 (1,782,010 ha), and 2014 (1,819,188 ha). In 2016, maize was harvested from 1,010,097 ha, with a total production of 73,767,371 t. The average yield in 2016 was 7.3 t/ha, which was higher than in 2015 (5.4 t/ha), and slightly lower in comparison with 2014 (7.5 t/ha) [25]. When compared to 2016, the total production of maize decreased by 45.5% in 2017, while the average yield was only 4.0 t/ha [26]. In 2018, expected production of maize was 6,965,000 t, which was 73.3% higher than in 2017, with an average yield ofapproximately 7.6 t/ha [27]. High maize production in 2018 positioned Serbia among the top ten maize exporters [28] and among the top twenty maize producers in the world [29].

In considering these facts, the aim of this study was to determine the current state of the level of emerging *Fusarium* mycotoxins in Serbian maize. Additionally, an effort was made to relate the obtained results with the weather conditions recorded during the trial period.

## 2. Results

### 2.1. Occurrence of Emerging Toxins in Maize Samples

Maize samples collected during the 2016 harvest were analyzed and the results are shown in Table 1. MON and BEA had the highest presence among emerging mycotoxins (>80%), except in the West-Backa region (50%). Other emerging mycotoxins were not detected at all or were found in fewer samples (<20%). Overall, maize samples from the Middle-Banat region were the most contaminatedfor all investigated emerging mycotoxins. MON, BEA, and FUS were present in all regions. Mean levels of MON ranged from 189.97 µg/kg (West-Backa) to 920.10 µg/kg (Srem). BEA mean levels were between 6.82 µg/kg (West-Backa) and 34.79 µg/kg (Srem). FUS levels were the highest among all tested mycotoxins. They ranged from 328.50 µg/kg in South-Backa to 12,272.00 µg/kg in a sample from the West-Backa region. ENs were found in all regions except Srem, with the highest mean levels in samples originating from Middle-Banat.

The results of maize samples collected during 2017 are summarized in Table 2. ENs were not found in any of the four regions. MON, BEA, and FUS were found in all regions, except for FUS, which was not foundin the sample from North-Backa. All the samples from the South-Banat region were contaminated with MON and BEA. The highest mean levels of MON (499.00 µg/kg) and FUS (3415.88 µg/kg) were recorded in the West-Backa region, while the highest mean level of BEA (12.26 µg/kg) was recorded in the South-Banat region.

In 2018, MON was found in all three regions (South-Backa, North-Backa, and South-Banat), BEA and FUS were not found in samples from the South-Banat region and ENs were not present in any sample (Table 3). The mean levels of all tested mycotoxins were highest in the South-Backa region (MON 199.32 µg/kg, BEA 4.89 µg/kg and FUS 5793.79 µg/kg).

### 2.2. Climate Conditions

Reports from the Republic Hydrometeorological Service of Serbia [30] showed that the vegetation period of 2016 (April–September) in the territory of Serbia was warmer with somewhat higher precipitation than the long-term average. The deviation of mean daily temperatures during the vegetation period showed positive values (0.8 °C to 1.6 °C). The standardized precipitation index (SPI-3), determined for the summer period from 1 June to 31 August, showed normal humidity conditions for most of the territory of Vojvodina. However, in some parts of Vojvodina, moderate to extremely humid conditions were recorded (Figure 1B). Such conditions were registered in the Middle-Banat and West-Backa regions, and some parts of the South-Backa and South-Banat regions. In some production areas, strong winds and hail storms were recorded and certainly contributed to the damage of the grains and the occurrence of fungi on crops. The moisture in deeper soil layers in the middle of June was significantly reduced in Vojvodina as a result of a weaker inflow of precipitation in these areas.

If the weather conditions were observed in more detail, warm but unstable weather prevailed during the transition from May to June, and the agrometeorological conditions allowed the intensive development of maize. The trend of variable but warm weather continued in June. In the middle of the month, due to the influx of very hot air, the temperatures were considerably above the average for this period of the year. Maximum daily temperatures reached 36 °C on some days [30]. Thermal conditions were favorable for the intensive development of spring agricultural crops. By the end of the first decade of August, the weather was mostly dry and stable, but since the beginning of the second decade of August, the air temperatures moved around and were below average values. Maximum air temperatures were up to 28 °C, while the minimum morning temperatures were significantly below the average values for this time of the year [30]. Significant precipitation, mostly rain showers, was recorded on the territory of the entire country. During July and August, 2 to 3 times more rain was registered in the territory of Serbia compared to the average quantities.

Detailed data on precipitation and temperature obtained from Metos® automatic weather stations (Metos®, Pessl Instruments, Weiz, Austria) in the observed regions in 2016 were compared to the multiannual average for 1981–2010 [33].Precipitation data (Figure 2) show that the Middle-Banat region, which overall was the most contaminated with emerging *Fusarium* toxins, and Srem, which showed the highest values of toxins, had precipitation values higher than the average in June, but they were noticeably lower in July. Average daily air temperature data (Figure 2) show that the temperature was around or slightly above the long-term average until August, when the temperature decreased. Average daily air temperatures were similar in all regions.

Reports from the Republic Hydrometeorological Service of Serbia [31] showed that the vegetation period of 2017 (April–September) was warmer and dryer than the multiannual average. The mean daily temperatures were 0.9–1.7 °C higher than the average, while precipitation was 20% lower than the average. SPI-3 showed extreme drought in the South-Banat region, while other observed regions were affected by high to moderate drought (Figure 1C).

The vegetation period started with unusually cold weather in April, but the weather conditions quickly normalized and became optimal for plant development during May [31]. During June, mean daily temperatures were higher than the average [31]. Maximum daily temperatures reached over 35 °C on some days, especially during the first and the last decade of the month. Hot weather continued during July. In most regions, precipitation was below the multiannual average, especially in the South-Banat region, where it was 50% lower than the average. The temperatures at the beginning of August were extremely high (38–42 °C) and higher than the average during the whole month [31]. Most regions were affected with drought, except the South-Banat region, which had 50% more rainfall than the average.

Precipitation and average daily air temperature data obtained from Metos® automatic weather stations (Metos®, Pessl Instruments, Weiz, Austria) in the observed regions during 2017 were compared to the multiannual average from 1981–2010 [33]. Precipitation data (Figure 3) showed that all regions had lower precipitation than the average during June. This trend continued through July for all regions except West-Backa, where high occurrences of MON and FUS were observed. The deviation of average daily air temperature (Figure 3) shows that temperature in April was lower than the multiannualaverage in all regions. During May, it was slightly above the multiannualaverage in all regions except South-Banat. In June the temperature was 1.4–2.4 °C higher than the average. Temperatures continued to be above the average until September.

According to reports from the Republic Hydrometeorological Service of Serbia [32], the vegetation period in 2018 (April–September) was 1.8–2.6 °C warmer than the multiannual average. SPI-3 showed normal to moderately humid weather conditions in all observed regions (Figure 1D).

The vegetation period started with unusually warm weather in April when the mean daily temperature was 4–5 °C above the multiannualaverage [32]. During May, weather conditions were optimal for plant growth. In the beginning of June, maximum daily temperatures were high (28–34 °C), but temperature decreased in the last decade of the month when maximum daily temperatures were in the range from 19 °C to 24 °C [32]. Frequent rain showers were recorded during June and continued through July. Warm weather with temperatures above multiannualaverage continued through August and September [32].

The deviation of precipitation and average daily air temperature data collected from Metos® automatic weather stations (Metos®, Pessl Instruments, Weiz, Austria) during 2018 from the multiannual average of 1981–2010 [33] was observed. Precipitation data (Figure 4) showed that precipitation was below average during the whole vegetation period in the North-Backa region. Precipitation was lower than the average in all three regions in April, May, August, and September. In June, precipitation was above average in the South-Backa and South-Banat regions. South-Backa was the only region where high precipitation continued in July.Average daily air temperature data (Figure 4) show that the temperature was above the long-term average in all regions during April and May. Temperature decreased during June and in July, reaching the multiannualaverage value in the South-Backa region, while the North-Backa and South-Banat regionswere below average. In August and September, the temperature increased again above the multiannual average in all observed regions.

### 2.3. Statistical Analysis

The Kruskal–Wallis test (95% confidence level) found significant differences in MON levels deriving from different years and different regions. Additionally, MON levels significantly differed when compared in terms of average seasonal temperature and precipitation. For FUS, significant differences were found between years, average seasonal temperatures and precipitation, but not between regions. Statistically significant differences in BEA levels were found between years and regions. Furthermore, the Spearman correlation determined a slightly moderate negative linear correlation between average seasonal temperatures and mycotoxin contamination levels (*r* = −0.41 for MON, *r* = −0.5 for BEA, and *r* = −0.45 for FUS). For monthly weather data, only temperatures in May showed a moderate negative linear correlation (*r* = −0.5 for MON, r = −0.58 for BEA, and r = −0.48 for FUS), while precipitation in May showed a moderate positive linear correlation with contamination levels of observed mycotoxins (*r* = 0.51 for MON, *r* = 0.59 for BEA, and *r* = 0.49 for FUS). Stepwise regression showed (with 95% confidence level) that air temperatures from May to August had statistically significant differences in MON levels. For FUS, temperatures and precipitation in any month, as well as average values for the whole growing season, did not show statistically significant differences among contamination levels. May temperatures had a statistically significant difference in BEA levels. The adjusted R-squared values were between 0.101 and 0.2278 and showed that these models cannot be used as prediction models; however, they gave insight about statistically significant variables that influence mycotoxin contamination.

## 3. Discussion

Sutton [34] explained that for maize, *Fusarium* infection of the ear most frequently takes place through the tip of the ear when the fungi penetrate through the silk during maize flowering. Very humid weather during the period from silking to ripening enables ear contamination [35]. The ear is the most susceptible to contamination at the beginning of silking, while the susceptibility decreases with silk aging [36,37]. The silking period in the climatic region of Serbia takes place during July and the first half of August.

According to the Republic Hydrometeorological Service of Serbia [30], most of the critical period for *Fusarium* infection of maize (July–August) in 2016 was characterized as dry and stable weather. However, since the beginning of the second decade of August, air temperatures ranged around and below average values [30]. Maximum air temperatures were up to a maximum of 28 °C, while the minimum morning temperatures were significantly below the average values for this time of the year. However, moderate to extremely humid conditions occurred during the summer of 2016 in the Middle-Banat and West-Backa regions and some parts of the South-Backa and South-Banat regions, which may have led to *Fusarium* fungi growth and the consequent production of mycotoxins. Moreover, cool, cloudy and humid weather during July and August did not favor agricultural crops and such conditions probably caused plant stress and higher susceptibility to *Fusarium* infection.

In 2017, precipitation was lower than the multiannual average during summer months in all regions except South-Banat, where high levels of BEA were recorded. High precipitation in these two regions during silking (July and beginning of August), when the maize is the most susceptible to *Fusarium* infections were favorable for fungal development, may have led to high levels of emerging toxins in samples from these regions. 

High precipitation during June 2018 in the South-Backa and South-Banat regions and during July 2018 in South-Backa resulted in South-Backa having the highest mean levels of MON, BEA, and FUS. Warm weather during July, together with high humidity in the South-Backa region enabled *Fusarium* infection of the ears, which may be related to the high mycotoxin contamination of samples from this region.

Emerging fusariotoxins were mostly investigated in Mediterranean countries. Juan et al. [22] analyzed 93 samples of organic cereals and organic cereal products from several local markets in Italy for the presence of different mycotoxins, including BEA, EN A, EN A1, EN B, EN B1. The authors found that levels of some emerging *Fusarium* mycotoxins ranged as follows: BEA 6.7–41 µg/kg, EN A 7.2–29.8 µg/kg, EN A1 5.3–64.3 µg/kg, EN B 5.5–102 µg/kg and EN B1 5.5–33.1 µg/kg. Among the commodities, the occurrence was the highest in wheat samples. Serrano et al. [21], investigated fusariotoxins’ occurrence in the Mediterranean area. They found that BEA was present in 2 of 14 maize samples and in 1 of 22 maize-based products. Obtained levels were 2.1 and 73.9 µg/kg for maize and 5.2 µg/kg for maize-based products, respectively. A high divergence among detected BEA levels was also found in this study (0.03–136 μg/kg). Later, Serrano et al. compared levels of emerging fusariotoxins between organic and conventional pasta in Spain [38]. They found that organic pasta was more contaminated than the conventional type. Contamination levels were 0.10–20.96 μg/kg for BEA, and 0.05–8.02 μg/kg for FUS, while ENs levels were 0.25–979.56 μg/kg. Remarkably high levels of emerging mycotoxins in raw cereals were found in Morocco by Zinedine et al. [39]. EN A1 was predominant among ENs with a presence in 39% of samples and levels ranging from 14 to 445 mg/kg. BEA was found in 26.5% of samples, with levels ranging from 1 to 59 mg/kg, while FUS was present in 7.8% of samples (levels from 0.6 to 2 mg/kg). Regarding maize samples, 42% contained ENs with mean levels of 207 mg/kg (EN A1), 54 mg/kg (EN B), 8 mg/kg (EN B1), while EN A was not detected. A similar situation was observed in cereals from the Spanish market [40]. The authors reported very a high presence of ENs (73.4%), wherein EN A1 was the most frequent with the highest levels (33.36–814.42 mg/kg). BEA was found in 32.8% of samples in the range of 0.51–11.78 mg/kg, and FUS levels were between 1.01–6.63 mg/kg with the presence in 7.8% of samples. In maize samples, the presence of ENs was 89%, BEA was found in 21% and FUS was in only one sample; on the other hand, ENs were found only during 2016 in this study, but not in any samples from 2017 and 2018, while BEA and FUS were detected every year. The highest mean level in maize was obtained for EN A1 of 813.01 mg/kg, while in this survey the highest mean level of EN A1 was only 9.30 μg/kg. In another study in Morocco on maize-based breakfast cereals, Mahnine et al. [41] obtained mean levels of 113 mg/kg for EN A1 and 20.1 mg/kg for EN B1, respectively, while EN A, EN B, FUS, and BEA were below LOQ in all samples. Tunisian cereals were highly contaminated as well. Maize-based cereals only contained EN A1 and ENB1 with mean levels of 113 mg/kg and 20.1 mg/kg. Oueslati et al. [20] obtained the presence of ENs in 96% of samples, where once more EN A1 was predominant (92.1%). Mean values were the highest in the case of EN A1 (up to 480 mg/kg). Only 3 samples of maize were analyzed. Two samples were positive, one with EN A1 (29.6 mg/kg) and another contained EN B1 (17.0 mg/kg). Notably, none of these authors analyzed results along with the weather conditions. Emerging fusariotoxins were also studied in rice. In Morocco, considerable contamination with ENs and BEA was revealed, but not with FUS [19]. In rice samples from Iran, a significant presence was found only in the case of BEA (40%), but in very low amounts [23].

Emerging fusariotoxins were also studied in some non-Mediterranean European countries. Goertz et al. [42] investigated the contamination of different maize hybrids in Germany during 2006 and 2007. BEA was found in 52% of samples from 2006 and in 33% of samples from 2007. Mean levels were 390 μg/kg and 240 μg/kg, respectively. MON was detected in 45% and 43% of samples, respectively, with mean values of 280 μg/kg and 110 μg/kg, respectively. Among EN, only EN B was investigated. Although its presence was relatively high (41% and 30%, respectively) the levels were the lowest (mean of 70 μg/kg and 160 μg/kg, respectively) among investigated emerging fusariotoxins. The authors explained that moderate temperatures and frequent precipitation recorded during early growth stages in 2007 were favorable for *Fusarium* growth. This is in accordance with the results of correlation analysis in this study, which showed a moderately negative correlation between May temperatures and toxin contamination, together with a moderately positive correlation between May precipitation and toxin contamination. However, a higher mycotoxin presence and higher contamination levels in Germany were found in samples from 2006, which they associated with maize exposure to drought stress in July and September 2006. In Norway, Uhlig et al. [43] investigated MON occurrence in Norwegian grain (oats, barley, and wheat) during a three-year period (2000–2002). MON was found in 46% of samples and the obtained levels were between 43 and 950 μg/kg. The authors noted that the highest prevalence of MON was found in the 2002 season (67%), along with the highest concentration. In Poland, Chelkowski et al. [44] detected ENs and BEA in 18 out of 27 maize samples (levels of 0.8–46.0 mg/kg). Unfortunately, in both studies, the weather conditions were not discussed. 

To the best of our knowledge, studies on emerging *Fusarium* toxins in Serbia have not been done to date. On the other hand, some studies have occurred in surrounding countries. In Romania, Stanciu et al. [45] found that ENs were the most frequent (73%) mycotoxins in both wheat and wheat flour, while EN B was detected the most (71%). The highest observed concentration was 407 µg/kg in wheat samples. Mean values were 19 µg/kg in wheat flour and 128 µg/kg in wheat. In neighboring Croatia, Jurjević et al. [46] investigated BEA presence in 209 maize samples originating from the 1996 and 1997 growing seasons. The authors found that 17.4% of samples from 1996 contained BEA at the mean level of 393 µg/kg and maximum concentration of 1864 µg/kg. In samples from 1994, only one of 104 samples contained BEA.

Based on the obtained results and available published data, the results from this study are in accordance with those found in Croatia, Italy, and Germany, while results from Poland, Spain, Morocco, and Tunisia are one order of magnitude higher. Unfortunately, studies from Romania, Norway, and Iran did not include maize or maize-based products and therefore a valid comparison cannot be made.

## 4. Conclusions

The main source of emerging *Fusarium* mycotoxins are cereals that are used in food and feed production, and they may thus pose a potential risk for human and animal health. Since environmental conditions in Serbia are favorable for the occurrence of mycotoxigenic fungi, monitoring of “traditional” but also “emerging” *Fusarium* toxins is essential for producing safe food and feed. The results indicated that most attention should be paid to fusaproliferin (FUS) and moniliformin (MON). Additionally, monitoring studies for emerging *Fusarium* mycotoxins are necessary for legislative purposes, because in the near future appropriate maximum contamination levels should be set for several mycotoxins by relevant authorities [38].

## 5. Materials and Methods

### 5.1. Samples

In total, 190 representative samples from commercial fields in Serbia were analyzed. Samples were collected during harvest in the northern Serbian province of Vojvodina, which is the country’s most important agricultural area, over three years: 73 samples from 28 localities in 2016, 72 samples from 12 localities in 2017 and 45 samples from 13 localities in 2018. Localities were clustered based on their administrative area into 6 regions: West-Backa, South-Backa, Srem, Middle-Banat, South-Banat, and North-Backa (Figure 1A).

Each sample was transported to the laboratory immediately after sampling and stored in a freezer at −20 °C until analysis. Prior to analysis, the samples were allowed to reach room temperature. All samples were milled on a laboratory mill so that >93% passed through a sieve with a pore diameter of 0.8 mm and a portion was taken for analysis.

### 5.2. Extraction and Mycotoxin Analysis in Maize Samples

Five grams of each milled sample were extracted using a 20 mL extraction solvent (acetonitrile–water–acetic acid (VWR, Vienna, Austria), 79:20:1, *v*/*v*/*v*) followed by a 1 + 1 dilution using acetonitrile–water–acetic acid (VWR, Vienna, Austria) (20:79:1, *v*/*v*/*v*) and a direct injection of 5 µL diluted extract. 

Liquid chromatography-tandem mass spectrometry (LC-MS/MS) screening of target fungal metabolites was performed at theInstitute of Bioanalytics and Agro-Metabolomics, Department of Agrobiotechnology (IFA-Tulln), University of Natural Resources and Life Sciences, Vienna, with a QTrap 5500 LC-MS/MS System (Applied Biosystems, Foster City, CA, USA) equipped with a TurboIon Spray electrospray ionization (ESI) source and a 1290 Series HPLC System (Agilent, Waldbronn, Germany). Chromatographic separation was performed at 25 °C on a Gemini^®^ C_18_-column, 150 × 4.6 mm i.d., 5 µm particle size, equipped with a C_18_ 4 × 3 mm i.d. security guard cartridge (all from Phenomenex, Torrance, CA, USA). The chromatographic method, chromatographic and mass spectrometric parameters, as well as the method validation data, are described by Malachova et al. [47]. Electrospray ionization-tandem mass spectrometry (ESI-MS/MS) was performed in the time-scheduled multiple reaction monitoring (MRM) mode both in positive and negative polarities in two separate chromatographic runs per sample by scanning two fragmentation reactions per analyte. The MRM detection window of each analyte was set to its expected retention time of ±27 sec and ±48 sec in the positive and the negative mode, respectively. Confirmation of a positive analyte identification was obtained by the acquisition of two MRMs per analyte (excepting moniliformin, which exhibits only one fragment ion), which yielded 4.0 identification points according to commission decision 2002/657/EC. In addition, the LC retention time and the intensity ratio of the two MRM transitionswas in accordance with the related values of an authentic standard within 0.1 min and 30% rel., respectively.

Quantification was based on an external calibration using a serial dilution of a multianalyte stock solution, and results were corrected for apparent recoveries. The accuracy of the method is verified on a continuous basis by regular participation in proficiency testing schemes [47,48] organized by BIPEA (Gennevilliers, France). Based on the submitted results, a general expanded measurement uncertainty of 50% has been determined [49]. In the case of the 175 results already submitted for maize and maize-based feed, 168 results were in the satisfactory range (*z*-score between −2 and 2).

### 5.3. Statistical Analysis

Statistical analysis (Appendix A) was performed using the computing environment *R* (*R* Core Team, Vienna, Austria) [50] on the data from regions where samples were collected during all three years of research (South-Backa and South-Banat). The Shapiro–Wilk normality test was used to check the distribution of the data. Since the data were not normally distributed, nonparametric tests were used for further analysis. The Kruskal–Wallis test was used to check whether the mean ranks of the mycotoxin levels were the same in all groups. Spearman’s correlation was used to determine the correlation between climate conditions and mycotoxin contamination levels. Furthermore, Stepwise regression with backward steps was used to obtain the optimal model and significant months in terms of temperature and precipitation values that influence mycotoxin levels.

## Figures and Tables

**Figure 1 toxins-11-00357-f001:**
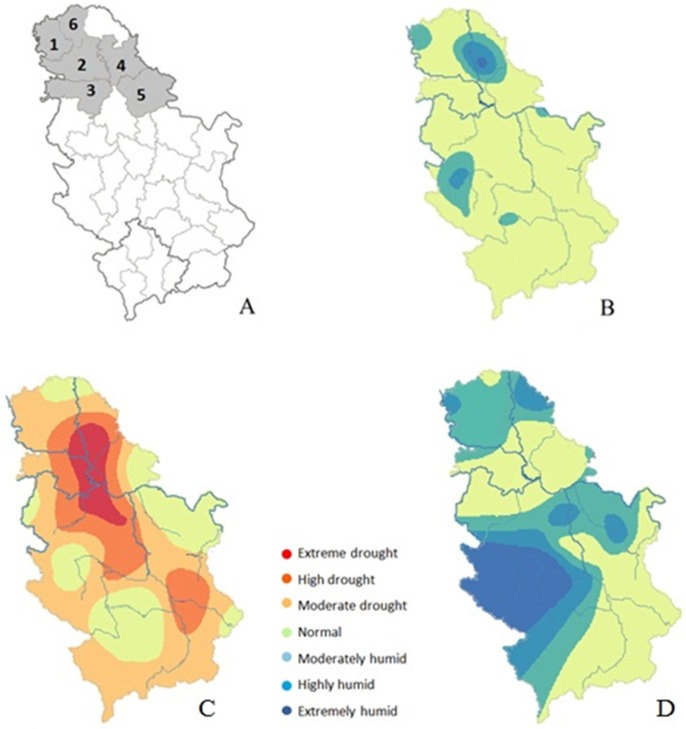
Regions of sample origin: **1**—West-Backa, **2**—South-Backa, **3**—Srem, **4**—Middle-Banat, **5**—South-Banat, **6**—North-Backa. (**A**) Humidity conditions in Serbia based on the Standardized Precipitation Index (SPI-3) determined for the summer period from 1 June to 31 August in 2016 (**B**) Reproduced from Agrometeorološki uslovi u proizvodnoj 2015/2016 godini, 2017, Republic Hydrometeorological Service of Serbia [30], 2017 (**C**) Reproduced from Agrometeorološki uslovi u proizvodnoj 2016/2017 godini, 2018, Republic Hydrometeorological Service of Serbia [31], and 2018 (**D**) Reproduced from Agrometeorološki uslovi u proizvodnoj 2017/2018, 2019, Republic Hydrometeorological Service of Serbia [32].

**Figure 2 toxins-11-00357-f002:**
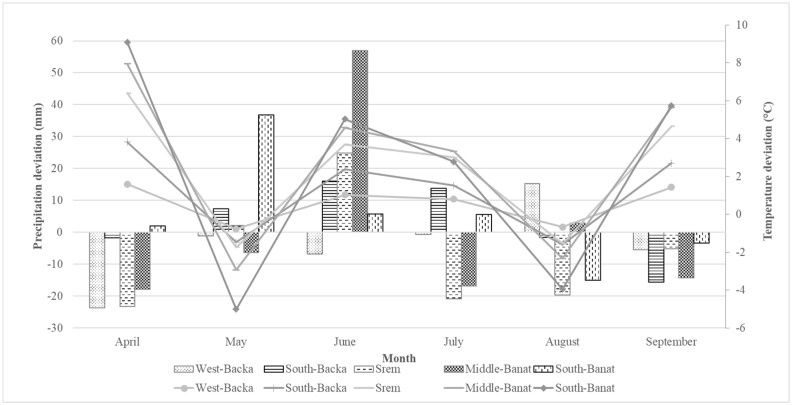
Deviation of total rainfall amount (columns) and average daily air temperature (lines) from the multiannual average (1981–2010) in 2016.

**Figure 3 toxins-11-00357-f003:**
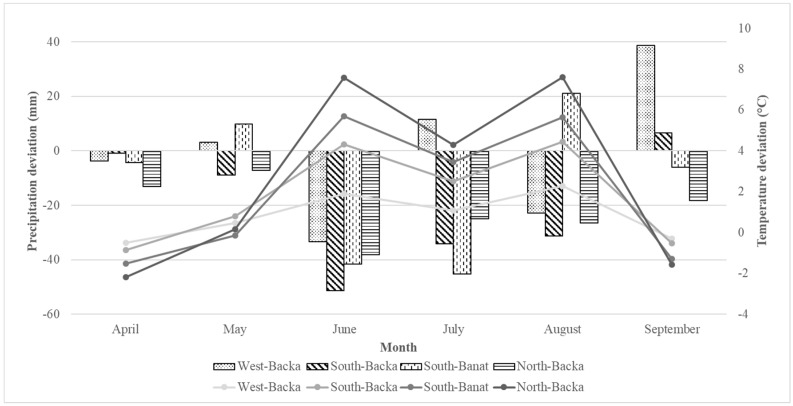
Deviation of total rainfall amount (columns) and average daily air temperature (lines) from multiannual average (1981–2010) in 2017.

**Figure 4 toxins-11-00357-f004:**
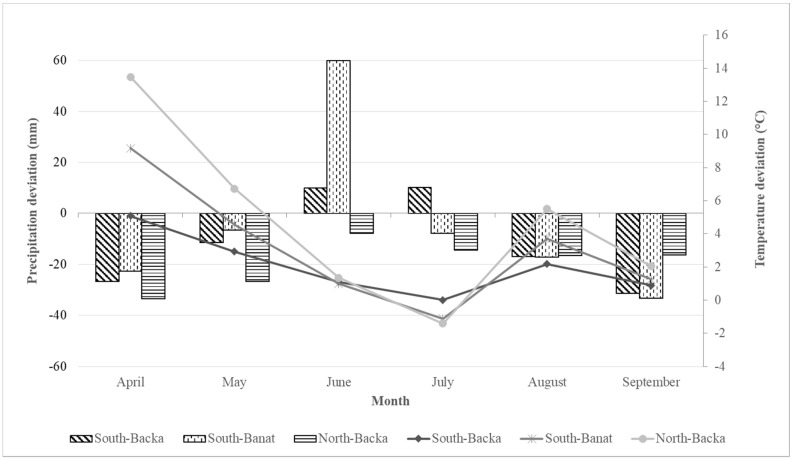
Deviation of total rainfall amount (columns) and average daily air temperature (lines) from the multiannual average (1981–2010) in 2018.

**Table 1 toxins-11-00357-t001:** Occurrence of emerging toxins in maize samples collected in the Republic of Serbia in 2016.

	MON	BEA	EN A	EN A1	EN B	EN B1	FUS
**South-Banat region**							
Average ± SD (µg/kg)	237 ± 230	18.7 ± 30.3	-	0.59	7.55	4.86	328 ± 268
Range (µg/kg)	5.06–850	0.41–129	-	-	-	-	85.4–1121
Samples	21	21	21	21	21	21	21
Positive samples (%)	21 (100.0)	20 (95.2)	0	1 (4.8)	1 (4.8)	1 (4.8)	16 (76.2)
**South-Backa region**							
Average ± SD (µg/kg)	534 ± 410	13.7 ± 26.7	0.25 ± 0.19	0.24 ± 0.18	-	-	827 ± 1032
Range (µg/kg)	15.3–1450	0.10–111	0.12–0.47	0.13–0.44	-	-	91.3–4687
Samples	29	29	29	29	29	29	29
Positive samples (%)	26 (89.7)	26 (89.7)	3 (10.3)	3 (10.3)	0	0	22 (75.9)
**Middle-Banat region**							
Average ± SD (µg/kg)	576 ± 391	7.06 ± 14.4	8.78 ± 11.8	9.30 ± 15.7	0.80 ± 1.01	8.27 ± 11.4	1018 ± 396
Range (µg/kg)	7.18–1228	0.23–49.7	0.41–17.1	0.11–27.4	0.08–1.52	0.20–16.3	450–1738
Samples	12	12	12	12	12	12	12
Positive samples (%)	12 (100.0)	11 (91.7)	2 (16.7)	3 (25.0)	2 (16.7)	2 (16.7)	11 (91.7)
**Srem region**							
Average ± SD (µg/kg)	920 ± 1649	34.8 ± 67.8	-	-	-	-	1736 ± 2384
Range (µg/kg)	3.03–3856	0.27–136	-	-	-	-	312–4488
Samples	5	5	5	5	5	5	5
Positive samples (%)	5 (100.0)	4 (80.0)	0	0	0	0	3 (60.0)
**West-Backa region**							
Average ± SD (µg/kg)	190 ± 192	6.82 ± 9.90	0.49	0.53	-	0.22	12272
Range (µg/kg)	34.8–405	0.03–18.2	-	-	-	-	-
Samples	6	6	6	6	6	6	6
Positive samples (%)	3 (50.0)	3 (50.0)	1 (16.7)	1 (16.7)	0	1 (16.7)	1 (16.7)

**Table 2 toxins-11-00357-t002:** Occurrence of emerging toxins in maize samples collected in the Republic of Serbia in 2017.

	MON	BEA	EN A	EN A1	EN B	EN B1	FUS
**South-Banat region**							
Average ± SD (µg/kg)	404 ± 493	12.3 ± 18.9	-	-	-	-	353 ± 414
Range (µg/kg)	10.4–1803	0.22–67.4	-	-	-	-	63.1–1275
Samples	17	17	17	17	17	17	17
Positive samples (%)	17 (100.0)	17 (100.0)	0	0	0	0	10 (58.8)
**South-Backa region**							
Average ± SD (µg/kg)	179 ± 255	6.30 ± 16.4	-	-	-	-	468 ± 843
Range (µg/kg)	2.68–1071	0.04–75.9	-	-	-	-	45.5–3018
Samples	33	33	33	33	33	33	33
Positive samples (%)	20 (60.6)	21 (63.6)	0	0	0	0	12 (36.4)
**West-Backa region**							
Average ± SD (µg/kg)	499 ± 880	1.96 ± 3.88	-	-	-	-	3416 ± 9786
Range (µg/kg)	1.66–2999	0.06–13.4	-	-	-	-	40.6–29512
Samples	21	21	21	21	21	21	21
Positive samples (%)	12 (57.1)	11 (52.4)	0	0	0	0	9 (42.9)
**North-Backa region**							
Average ± SD (µg/kg)	221	18.3	-	-	-	-	-
Range (µg/kg)	-	-	-	-	-	-	-
Samples	1	1	1	1	1	1	1
Positive samples (%)	1 (100.0)	1 (100.0)	0	0	0	0	0

**Table 3 toxins-11-00357-t003:** Occurrence of emerging toxins in maize samples collected in the Republic of Serbia in 2018.

	MON	BEA	EN A	EN A1	EN B	EN B1	FUS
**South-Banat region**							
Average ± SD (µg/kg)	39.5 ± 44.6	-	-	-	-	-	-
Range (µg/kg)	5.19–89.9	-	-	-	-	-	-
Samples	6	6	6	17	6	6	6
Positive samples (%)	3 (50.0)	0	0	0	0	0	0
**South-Backa region**							
Average ± SD (µg/kg)	199 ± 238	4.89 ± 6.91	-	-	-	-	5794 ± 14,479
xRange (µg/kg)	5.80–857	0.15–21.5	-	-	-	-	63.2–38,610
Samples	34	34	34	34	34	34	34
Positive samples (%)	16 (47.1)	9 (26.5)	0	0	0	0	7 (20.6)
**North-Backa region**							
Average ± SD (µg/kg)	34.4 ± 36.9	3.33	-	-	-	-	72.4
Range (µg/kg)	55.82–88.7	-	-	-	-	-	-
Samples	5	5	5	5	5	5	5
Positive samples (%)	4 (80.0)	1 (20.0)	0	0	0	0	1 (20.0)

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
