# Peer review of "Emerging Fusarium Mycotoxins Fusaproliferin, Beauvericin, Enniatins, and Moniliformin in Serbian Maize"

_toxins, 2019, doi:10.3390/toxins11060357_

Reviewer 1 Report

The manuscript describes the mycotoxin content of Serbian maize. It is not original and has no research interest nowadays. I do not think it should be published in this journal.

Author Response

Point 1: The manuscript describes the mycotoxin content of Serbian maize. It is not original and has no research interest nowadays. I do not think it should be published in this journal.

Response 1: In this case, we investigated the presence of emerging Fusarium toxins: Fusaproliferin, Beauvericin, Enniatins, and Moniliformin. There are many works regarding major Fusarium toxins, such as deoxynivalenol and T-2 toxin, zearalenone, and fumonisin, but not yet about emerging ones

Reviewer 2 Report

The paper report the occurrence of “emerging Fusarium-mycotoxins” in some serbian maize, also considering the weather data. In my view, the paper need to a statistical analysis correlating the "mycotoxins" data with "climatic" data (considering the different years), also in order to improve the discussion section.

In the subsection “2.2 Climate conditions”, the authors report some sentences that are more appropriate in the discussion section.  Additionally, in that section, the data mentioned in the text, not always are referred to a figure or a table.

The data of rainfall and temperature for each year can be showed in the same chart, using secondary axes (figures: 2 and 3; 4 and 5; 6 and 7), in order to facilitate the comprehension of the data.

In the discussion section, although the authors comment exhaustively the data reported in other studies, they are not comment them with those obtained with the present study.

I attach few correction below:

Please uniform the using of the term "Fusarium"throughout in the manuscript; not always, it is write in Italic.

In the figures 3, 5 and 7 on "y axex" use the point for decimal numbers.

Author Response

Response to Reviewer 2 Comments

Point 1: The paper report the occurrence of “emerging Fusarium-mycotoxins” in some serbian maize, also considering the weather data. In my view, the paper need to a statistical analysis correlating the "mycotoxins" data with "climatic" data (considering the different years), also in order to improve the discussion section.

Response 1: We performed correlation analysis (as well as some additional ones) in statistical program R. Results are summarized in the last part of the Results section (2.3. Statistical analysis), and we would like to include the code with row results in Appendix 1. Also, some changes are made in the discussion section.

Point 2 and 3: In the subsection “2.2 Climate conditions”, the authors report some sentences that are more appropriate in the discussion section.  Additionally, in that section, the data mentioned in the text, not always are referred to a figure or a table.

The data of rainfall and temperature for each year can be showed in the same chart, using secondary axes (figures: 2 and 3; 4 and 5; 6 and 7), in order to facilitate the comprehension of the data.

Response 2 and 3: We accepted your suggestion and merged charts for rainfall and temperature for each year. We renamed the charts and properly referred them in the text. Regarding some sentences from the Results section that would fit better in Discussion, can you, please, point them specifically? And we will make necessary adjustments.

Point 4: In the discussion section, although the authors comment exhaustively the data reported in other studies, they are not comment them with those obtained with the present study.

Response 4: We made some connections between the data of this and other studies. However, it is a challenge to make parallels when there are not so many data from other studies for maize, we mentioned levels of contamination in other products.

Point 5: Please uniform the using of the term "Fusarium"throughout in the manuscript; not always, it is write in Italic.

Response 5: We corrected it, it is uniform now (Fusarium), thank you.

Point 6: In the figures 3, 5 and 7 on "y axex" use the point for decimal numbers.

Response 6: We corrected it, thank you.

Reviewer 3 Report

This paper describes the study on emerging Fusarium mycotoxin including BEA, ENs, MON and FUA in Servia. The authors monitored the levels of these emerging Fusarium mycotoxins in maize and evaluated the mycotoxin levels in accordance with environmental conditions in Servia. The experimental treatments were well designed and the results were described properly. The manuscript can be evaluated as academically valuable to increase food safety, because there have been limited data related to emerging mycotoxins such as BEA, ENs, MON and FUS, so far. This manuscript can be accepted after “major revision”

Following major points should be considered for revision.

1. The authors discussed on the relationship of the levels of emerging Fusarium mycotoxins in maize with environmental conditions in Servia. All data related with environmental conditions are expressed only with average data without data range. All figures should be revised to see data range.

2. Method validation is one of the most important part of monitoring studies on mycotoxins. Even though the authors take part in the proficiency test continuously, the data on method validations such as accuracy, precision and recovery with the mass chromatogram of tested mycotoxins for this experiment should be included in text or as supplementary data.   

Author Response

Point 1: The authors discussed on the relationship of the levels of emerging Fusarium mycotoxins in maize with environmental conditions in Servia. All data related with environmental conditions are expressed only with average data without data range. All figures should be revised to see data range.

Response 1: Dear reviewer, thank you for your suggestion. However, we were asked by another reviewer to merge together graphs for temperature and precipitation of the same year. We did as advised and now those graphs would be difficult to understand if we add data range. Can you, please, check on them after this revision and give advice about them if needed. Thank you for your understanding.

Point 2: Method validation is one of the most important part of monitoring studies on mycotoxins. Even though the authors take part in the proficiency test continuously, the data on method validations such as accuracy, precision and recovery with the mass chromatogram of tested mycotoxins for this experiment should be included in text or as supplementary data.

Response 2: Validation data is published elsewhere, and we added the source (In the Methods section and References). Thank you for the advice.

Round  2

Reviewer 2 Report

Although the authors with statistical analysis have improved the manuscript, they still have to improve it in terms of writing and understanding.

I enclose a first version with related modifications, in order to facilitate the revision.

Author Response

Dear reviewer,

We are submitting a new version of our manuscript.

Your comments helped us to improve the Results and the Discussion section, as well as Methods regarding statistical analysis. We addressed them all.

Furthermore, our manuscript was checked by a native English speaker who helped us to improve the manuscript in terms of grammar, sentence structure and word choice.

On the other hand, we also sent the manuscript to Elsevier professional Language Editing Service, but their revision takes more time, and we are expecting their delivery by next Wednesday.

Best regards,

The authors

Reviewer 3 Report

The manuscript was well revised according to my previous revision requests and it can be accepted for the publication in "toxins" as a research article.

Author Response

Dear reviewer,

We are submitting a new version of our manuscript in which we addressed the comments from another reviewer.

In this version we improved the Results and the Discussion section, as well as Methods regarding statistical analysis.

Furthermore, our manuscript was checked by a native English speaker who helped us to improve the manuscript in terms of grammar, sentence structure and word choice.

On the other hand, we also sent the manuscript to Elsevier professional Language Editing Service, but their revision takes more time, and we are expecting their delivery by next Wednesday.

Best regards,

The authors

Round  3

Reviewer 2 Report

The authors have improved the manuscript according to the suggestion. In my view now it can be considered for publication.